# Cancer Vaccines for Genitourinary Tumors: Recent Progresses and Future Possibilities

**DOI:** 10.3390/vaccines9060623

**Published:** 2021-06-09

**Authors:** Brigida Anna Maiorano, Giovanni Schinzari, Davide Ciardiello, Maria Grazia Rodriquenz, Antonio Cisternino, Giampaolo Tortora, Evaristo Maiello

**Affiliations:** 1Oncology Unit, Foundation Casa Sollievo della Sofferenza IRCCS, 73013 San Giovanni Rotondo, Italy; davideciardiello@yahoo.it (D.C.); grazia.rodriquenz@gmail.com (M.G.R.); e.maiello@operapadrepio.it (E.M.); 2Department of Translational Medicine and Surgery, Catholic University of the Sacred Heart, 00168 Rome, Italy; giovanni.schinzari@policlinicogemelli.it (G.S.); giampaolo.tortora@policlinicogemelli.it (G.T.); 3Medical Oncology Unit, Comprehensive Cancer Center, Foundation A. Gemelli Policlinic IRCCS, 00168 Rome, Italy; 4Medical Oncology, Department of Precision Medicine, Luigi Vanvitelli University of Campania, 80131 Naples, Italy; 5Urology Unit, Foundation Casa Sollievo della Sofferenza IRCCS, 73013 San Giovanni Rotondo, Italy; antonio.cisternino@operapadrepio.it

**Keywords:** prostate cancer, renal cancer, urothelial cancer, vaccines, immunotherapy

## Abstract

Background: In the last years, many new treatment options have widened the therapeutic scenario of genitourinary malignancies. Immunotherapy has shown efficacy, especially in the urothelial and renal cell carcinomas, with no particular relevance in prostate cancer. However, despite the use of immune checkpoint inhibitors, there is still high morbidity and mortality among these neoplasms. Cancer vaccines represent another way to activate the immune system. We sought to summarize the most recent advances in vaccine therapy for genitourinary malignancies with this review. Methods: We searched PubMed, Embase and Cochrane Database for clinical trials conducted in the last ten years, focusing on cancer vaccines in the prostate, urothelial and renal cancer. Results: Various therapeutic vaccines, including DNA-based, RNA-based, peptide-based, dendritic cells, viral vectors and modified tumor cells, have been demonstrated to induce specific immune responses in a variable percentage of patients. However, these responses rarely corresponded to significant survival improvements. Conclusions: Further preclinical and clinical studies will improve the knowledge about cancer vaccines in genitourinary malignancies to optimize dosage, select targets with a driver role for tumor development and growth, and finally overcome resistance mechanisms. Combination strategies represent possibly more effective and long-lasting treatments.

## 1. Introduction

Immunotherapy has represented a breakthrough therapy for many cancer subtypes in the last years. Among genitourinary (GU) neoplasms, the urothelial carcinoma (UC) and the renal cell carcinoma (RCC) have benefitted mainly from immune checkpoint inhibitors (ICIs) both as single agents and in combination with other ICIs or tyrosine kinase inhibitors (TKIs) [1,2,3,4,5,6,7,8,9,10,11,12,13]. However, in prostate cancer (PCa), ICIs have shown limited efficacy primarily due to an immunologically ‘cold’ and immunosuppressive tumor microenvironment (TME) [14,15,16,17,18].

Improving immunotherapy efficacy requires combination therapies or different pharmacological approaches [19]. In fact, the two principal ways to enhance the immune system’s antitumor activity are blocking the immune-suppressive signals responsible for the decreased antitumor response (that is, how ICIs work) or stimulating the immune activation against specific tumor-associated antigens (TAAs). The latter is the mechanism used by anticancer vaccines, capable of triggering the immune response actively by administering antigens conjugated with co-stimulatory molecules or loaded on patients’ immune cells [20,21,22,23]. In this way, antigen-presenting cells (APCs) can recognize, uptake, process, and present TAAs to naïve T-cells. Generally, intracellular antigens are presented with the class I major histocompatibility complex (MHC) molecules to CD8^+^ cells, turning them into effector cytotoxic lymphocytes (CTLs) [24]. It is more difficult to elicit a cytotoxic response in the case of extracellular antigens. The class II MHC molecules usually present them to CD4^+^ cells [24,25]. However, APCs—especially dendritic cells (DCs)—can process and present some extracellular antigens through the class I MHC to CD8^+^ cells, a process known as antigen cross-presentation whose discovery has been of great importance for therapeutic vaccines development [25].

Of note, the first two cancer vaccines with therapeutic use were approved for GU malignancies: Bacillus Calmette–Guerin (BCG) for non-muscle invasive bladder cancer (NMIBC) and Sipuleucel-T (Provenge^®^) for metastatic castration-resistant prostate cancer (mCRPC) [26,27]. Since then, many studies have been conducted, mainly in the metastatic setting of GU cancers, but no new approvals have followed due to unsatisfactory results.

We conducted a review to summarize the recent advances in using vaccines for GU malignancies treatment to find out their strengths and weaknesses for future applications.

## 2. Materials and Methods

We performed a literature search for papers reporting the clinical use of vaccines in neoplasms of the GU tract published in the last ten years (up to March 2021). We searched PubMed, Embase, and Web of Science, using keywords, including (‘vaccines’ or ‘vaccine therapy’) and (‘bladder cancer/carcinoma’ or ‘urothelial cancer/carcinoma’ or ‘kidney/renal cancer/carcinoma’ or ‘prostate cancer/carcinoma’ or ‘testicular/testis cancer/carcinoma’). We considered original researches published in peer-reviewed journals and conference abstracts in the English language from 2011 to 2021. We discarded letters, commentary, personal opinions. We included clinical trials (phases I–IV) enrolling human subjects, whereas we excluded studies on cellular and animal models.

A total of 61 studies were included in our review. No clinical use of vaccines in testicular cancer was published in the last ten years.

## 3. Results

Therapeutic cancer vaccines target TAAs alongside adjuvant molecules that can elicit specific antibodies or cytotoxic immune responses against cancer cells. There are different ways to present TAAs to the immune system. DNA and RNA encoding TAAs or whole peptides can be recognized and processed by the APCs; tumor cell lines express TAAs and can chemotactically attract APCs; viral vectors transfect APCs after being loaded with prespecified antigens; finally, DCs act as APCs and can be loaded with TAAs [20,21,22,23]. These different mechanisms have all been tested in PCa, RCC, and UC [23]. Once recognized, TAAs trigger APCs maturation. Subsequently, the interaction between class I MHC and complementary co-stimulatory ligands activates CD8^+^ T-cells with tumor-killing properties specifically targeting TAAs. Through class II MHC, APCs activate CD4^+^ T-lymphocytes. CD4^+^ can potentiate CD8^+^ T-lymphocytes proliferation and stimulate B-lymphocytes activation, resulting in specific antibodies production (Figure 1) [20,21,22,23].

### 3.1. Vaccine Therapy in Prostate Cancer (PCa)

PCa represents the most frequent tumor and the second leading cause of death among the Western male population [28]. PCa is an ideal candidate for vaccine therapies, given its high targetable number of TAAs, prostatic acid phosphatase (PAP), prostate-specific antigen (PSA) and prostate-specific membrane antigen (PSMA) among the most important [29,30]. The majority of studies focused on mCRPC. Only three phase III trials have been conducted. Even if specific immune activation was detectable, vaccines usually did not determine significant survival improvement (Table 1).

Hence, combination therapies and newer targetable antigens are under evaluation for improving vaccines efficacy (Table 2).

#### 3.1.1. Sipuleucel-T and PAP-Targeted Vaccines

PAP is an ideal candidate for vaccines, being expressed on the prostate epithelium [29,30]. Sipuleucel-T, a DCs vaccine loaded with PA2024 (PAP plus granulocyte-macrophage colony-stimulating factor (GM-CSF)), so far remains the only approved vaccine for asymptomatic or minimally symptomatic mCRPC patients. In the IMPACT phase III trial, Sipuleucel-T had yet improved overall survival (OS) (25.8 vs. 21.7 mos; HR = 0.78, 95% confidence interval [CI], 0.61–0.98; *p* = 0.03), but not progression-free survival (PFS), compared to placebo (PBO), with lower PSA levels at baseline predictive of higher OS (13.0 mos for PSA < 22.1 ng/mL, versus 2.8 mos for PSA > 134 ng/mL), as well as high antibodies production [26,31]. More recently, an attempt to combine Sipuleucel-T plus androgen deprivation therapy (ADT) was made in the STAND phase II clinical trial, that showed a higher humoral response, related to longer time to PSA progression (*p* = 0.007) in the non-metastatic castration-resistant prostate cancer (nmCRPC), when ADT followed Sipuleucel-T than vice versa [32]. Similarly, immune system activation was obtained when Sipuleucel-T was combined with the androgen receptor-targeted agent abiraterone in mCRPC patients. Of note, the immune response activation was not reduced by concomitant prednisone [33]. The rationale of these combinations stands on the capability of androgen-targeted agents of interfering with the immune system in various ways, for example favoring T-cell infiltration [29]. As previously attempted in the phase I study NCT01832870, another possible combination is Sipuleucel-T plus low-dose ipilimumab (1 mg/kg): 6/9 treated patients achieved >4 years OS [34,35]. In the mCRPC setting, Sipuleucel-T is currently under evaluation in association with ipilimumab (NCT01804465), Radium-223 (NCT02463799), and a glycosylated recombinant human interleukin (IL)-7 (NCT01881867). In the neoadjuvant setting, significant activation of T-cells in tumor biopsies was demonstrated with Sipuleucel-T before radical prostatectomy (RP). However, PFS and OS are not available [36].

DNA vaccines targeting PAP, such as pTVG-HP, induced a PSA decline in about 60% of mCRPC patients [37]. However, no metastasis-free survival (MFS) improvement was achieved if pTVG-HP was combined with pembrolizumab in recurrent PCa [38]. The combination of pTVG-HP with nivolumab in nmCRPC patients is currently under investigation (NCT03600350).

The use of multiple PAP-fused cytokines (human/mouse GM-CSF, IL-2, IL-4 and IL-7) represents a new strategy for vaccines efficacy to be enhanced [39].

#### 3.1.2. PROSTVAC and PSA-Based Vaccines

PSA is a classical biomarker for PCa diagnosis and disease monitoring, representing a promising vaccine candidate [29,30]. PROSTVAC (PSA-TRICOM) consists of two different poxviral vectors for human PSA (PROSTVAC-V and -F), plus three co-stimulatory molecules for T-cells (TRICOM). The phase III PROSPECT trial (NCT01322490), enrolling asymptomatic or minimally symptomatic chemotherapy-naïve mCRPC patients, did not show an OS improvement and was stopped early after meeting the futility criteria [40]. PSA-TRICOM has also been evaluated as intraprostatic administration, resulting in an increased CD4+ and CD8+ cells infiltrate in tumor biopsies, determining PSA stability in 10/19 patients [41,42]. The combination of PROSTVAC and ipilimumab was tested in a phase I trial in mCRPC patients, reporting a PSA decline in about half cases [43]. Baseline immune settings, such as lower PD1^+^, high Cytotoxic T-Lymphocyte Antigen (CTLA)-4^−^ Tregs, were associated with longer OS [44].

Multiple combination trials of PROSTVAC are ongoing: plus nivolumab before RP (NCT02933255, phase I/II), plus a neoantigen DNA vaccine, nivolumab and ipilimumab (NCT03532217, phase I) or plus docetaxel (NCT02649855, phase II) in metastatic hormone-sensitive prostate cancer (mHSPC), plus M7824 (a combined anti-programmed cell death protein-ligand 1 [PD-L1]/transforming-growth factor [TGF]-βR2 monoclonal antibody) and the recombinant Avipoxvirus vaccine CV301 in nmCRPC (NCT03315871, phase II).

Among the other PSA-targeted vaccines, the Listeria monocytogenes-based ADX31-142 is under evaluation combined with Pembrolizumab in the phase I/II study KEYNOTE-146 (NCT02325557).

#### 3.1.3. PSMA-Based Vaccines

PSMA is expressed on the PCa epithelium, representing an ideal candidate for vaccination [29,30]. In a phase I/II dose-escalation study, a DNA-based human leukocyte antigen (HLA)-A2 binding epitope from PSMA, fused to tetanus toxin, induced specific CD4^+^ and CD8^+^ T-cells in 32 nmCRPC patients, also increasing PSA-doubling time (DT) from 12 to 16.8 mos (*p* = 0.0417) [45]. The phase I/II trial NCT02514213 showed an 18-mos PFS rate of 85% with specific immune responses in over a quarter of nmCRPC patients after the INP-5150 DNA vaccine (double target of PSA and PSMA) [46]. In the CRPC setting, 21 patients were randomized to receive DC vaccines with recombinant PSMA and Survivin peptides versus docetaxel plus prednisone, reaching an ORR of 72.7% vs. 45.4% [47]. Instead, a viral replicon vector system (VRP) carrying PSMA did not induce clinical benefit, even if neutralizing antibodies were produced [48]. Among RNA vaccines, CV-9103 contains PSMA, PSA, prostate stem cell antigen (PSCA) and six-transmembrane epithelial antigen of the prostate-1 (STEAP1). In a phase I/II trial, CV-9103 induced a specific immune response. No survival data are available [49].

#### 3.1.4. Other TAAs and Personalized Peptide Vaccination (PPV)

Among the other peptides used as vaccine targets, minimal survival advantages besides immune responses have been detected (Table 1).

The androgen receptor (AR) is a crucial element for the proliferation and therapy of PCa [50]. In the phase I trial NCT02411786, the DNA vaccine pTGV-AR, targeting the androgen receptor (AR), induced longer PSA-PFS in 47% of treated mHSPC patients with higher T-cells activation (*p* = 0.003) [51]. The combination of pembrolizumab plus the double pTVG-HP/ pTVG-AR DNA vaccine is under evaluation (NCT04090528, phase II).

Mucin-1 (MUC1) is a glycoprotein expressed on epithelial cells’ apical surface [29]. In a phase I/II trial enrolling 17 patients with nmCRPC, MUC1 loaded-DCs improved PSA-DT (*p* = 0.037) [52]. More recently, a vaccine using the adenoviral vector Ad5 targeting PSA, MUC-1 and Brachyury induced a PSA decline in 2/12 mCRPC patients in a phase I trial [53].

NY-ESO-1 is a surface antigen expressed in 15–25% PCa cells [54]. In a phase I clinical trial, T-cell responses were detected in 9/12 stage IV patients [55]. Moreover, in a randomized phase IIa trial, 21 chemotherapy-naïve CRPC patients were vaccinated with DCs loaded with NY-ESO-1, melanoma-associated antigen (MAGE)-C2 and MUC1. Specific T-cells, detected in about 1/3 patients, correlated with radiological responses [56].

AE37, a human epidermal growth factor receptor (HER)-2 hybrid class I MHC peptide vaccine, was tested in a phase I trial and induced immunological responses and long memory (around 4 years) in case of subsequent vaccines boosters [57,58]. Predictive factors were also investigated: the presence of pre-existing immunity to the native peptide correlated with PFS; TGF-β inversely related to immunological responses and OS; delayed-type hypersensitivity was directly associated with OS; and HLA-A*24 and −DRB1*11 alleles induced more robust immunological responses and longer OS [57,58,59,60].

Cell division associated 1 (CDCA1) peptide vaccine, administered in the phase I trial NCT01225471 in 12 CRPC patients progressive to docetaxel, induced specific T-cells in a quarter of patients, reaching 11 mos of mOS [61]. UV1 peptide vaccine (targeting telomerase reverse transcriptase (TERT)) induced immune responses in 85.7% of mHSPC patients, with PSA declining in 64% of cases, in a phase I study [62]. The vaccination with T-cell receptor gamma chain alternate reading frame protein (TARP) and pulsed DCs induced specific immune responses and reduced PSA velocity in D0 PCa patients (NCT00908258) and is currently under evaluation as DCs vaccination in the phase II NCT02362451 study [63]. In a phase I/II trial (NCT03199872), a peptide vaccine against the Ras homolog gene family member C (RhoC) GTPase determined strong CD4+ responses in 18/21 patients after RP [64].

The NCT02390063 study demonstrated the elicitation of T-cell responses in patients with PCa, both before RP and during active surveillance, after the administration of two replication-deficient viruses, the ChAd (chimpanzee adenovirus) and the MVA (Modified Vaccinia Ankara) targeting 5T4, an oncofetal self-antigen [65]. The second one, called TroVax, was already tested with the good capability to induce immune responses in mCRPC and mRCC, being 5T4 a highly expressed epithelial antigen [66,67]. TroVax was also tested in a phase II trial combined with docetaxel, achieving a higher mPFS than docetaxel alone (9.67 vs. 5.1 mos, *p* = 0.097), strictly connected with baseline PSA [68,69]. Other possible combinations of docetaxel and vaccines have been tested: a Gleason score downstaging was evidenced in 4/6 patients undergoing RP after receiving docetaxel plus GVAX (genetically modified irradiated PCa cells), a vaccine initially administered in a negative phase III trial [70,71]. Similar evidence of immune response was obtained by using degarelix + cyclophosphamide + GVAX [72].

Personalized peptide vaccination (PPV) is a novel vaccine strategy consisting of the administration of selected HLA-matched peptides based on pre-vaccine immunity [73]. Despite PPV representing an attractive strategy for PCa, and even after positive phase II trials, no survival advantage over PBO emerged in the phase III trial enrolling 310 mCRPC patients progressing to docetaxel (HR = 1.04; 95%, CI 0.80–1.37; *p* = 0.77) [74,75,76,77]. Possible biomarkers with predictive roles have also been investigated. Haptoglobin levels were directly related to OS, whereas IL-6 levels were inversely associated with OS [76,78,79]. Moreover, in a post-hoc analysis of the phase III trial, a very low (<26%) or very high (>64%) proportion of lymphocytes at baseline determined an OS benefit [74]. However, the personalized vaccination strategy did not determine significant survival differences besides immune responses, even in combination with chemotherapy, as emerged in the treatment of CRPC with personalized autologous dendritic cell-based cancer vaccine (DCvac) plus docetaxel [80]. A phase III trial is currently ongoing comparing docetaxel + DCvac versus docetaxel + PBO in the first-line setting of mCRPC (NCT02111577).

Multi-peptides vaccines have also been tested in the CRPC setting, as they were thought to be more effective due to the immune induction against multiple targets. However, the 20-peptides vaccine KRM-20 determined only two partial responses (PR) and one PSA stability among 17 patients in a phase I trial [81]. Similarly, in a phase II trial, KRM-20 combined with docetaxel plus dexamethasone, even if increased specific antibodies and T-cells, did not determine differences in PSA decline and OS compared to PBO [82].

New peptides are under evaluation as vaccine targets: B-cell lymphoma extra-large protein (Bcl-xl)_42-CAF09b for mHSPC in the NCT03412786 phase I trial; the TENDU vaccine-targeting tetanus-epitope targeting (TET), in the phase I NCT04701021 for relapsing PCa after RP; the RV001V vaccine-targeting RhoC, in the phase II NCT04114825 for patients with biochemical recurrence after curative radiotherapy (RT) or RP; the combination of BN-Brachyury Vaccine, M7824, ALT-803 and Epacadostat (QuEST1-NCT03493945, phase I/II) in the mCRPC setting [83].

### 3.2. Vaccine Therapy in Urothelial Cancer (UC)

UC has a long and successful history of vaccines use, starting from the Bacillus Calmette–Guerin (BCG), which represents a cornerstone for the treatment of non-muscle invasive bladder cancer (NMIBC) since the 1990s [27]. However, BCG failure occurs in 20–50% of patients [84]. Intending to potentiate BCG efficacy, in a randomized phase I study (NCT01498172), 24 NMIBC patients received a vaccine containing the recombinant MAGE-A3 protein + the adjuvant AS15 before BCG instillations. In half of the patients, specific T-cells were subsequently detectable in blood. No survival data are available [85].

Some UC TAAs have been tested mainly on DCs or as peptide vaccines, inducing immune responses with controversial survival effects in phase I/II trials (Table 1). Survivin-2B80-88 improved OS in phase I (*p* = 0.0009) [86]. CDX-1307, targeting the mannose receptor, induced immune responses in bladder cancer (BCa), but it did not get over phase I because the N-ABLE NCT01094496 phase II study was stopped early due to slow enrollment [87]. DCs loaded with Wilms tumor (WT)-1 in seven patients with mUC or mRCC determined specific immune responses and decreased T-regs [88]. S-288310, derived from DEP domain-containing 1 (DEPDC1) and M-phase phosphoprotein 1 (MPHOSPH1), was administered to pretreated mUC patients in a phase I/II trial: 89% of patients developed specific T-cells, reaching mOS of 14.4 mos, with better results if a double induction against both peptides was obtained [89]. NEO-PV-01 peptide derives by the genomic profiling of patients’ BCa: in a phase Ib trial, 10/14 patients achieved PR or stable disease (SD) [90].

The PPV strategy has been evaluated in the platinum-progressing mUC. A phase I trial did not meet its primary endpoint of prolonging PFS among 80 BCa patients; however, a significantly longer OS was reached than best supportive care (7.9 vs. 4.1 mos; *p* = 0.049) [91]. In a phase II trial enrolling 48 patients with metastatic upper tract urothelial cancer (mUTUC), the development of specific T-cells was associated with longer OS (*p* = 0019). A median OS of 7.7 mos was achieved, reaching 13.0 mos if salvage chemotherapy was associated [92]. Finally, among 12 mUC patients, a phase I trial reported one complete response (CR), one PR, two SDs, mPFS of 3 mos, and mOS of 8.9 mos [93].

Current trials are ongoing: the peptide vaccine ARG1 (targeting arginase-1) is under evaluation as a single agent in a phase I trial (NCT03689192); the NCT03715985 study is evaluating the multi-peptides neo-antigen vaccine NeoPepVac in combination with anti-PD1/PD-L1 in many solid tumors, including mUC (Table 2).

### 3.3. Vaccine Therapy in Renal Cell Cancer (RCC)

In the mRCC, different DCs and peptide vaccines have been tested, mostly in phase I/II trials, with only two published phase III studies (Table 1). With a similar Sipuleucel-T mechanism, Rocapuldencel-T is composed of DCs plus amplified tumor RNA plus CD40L RNA. In the ADAPT phase III trial, 462 patients were randomized 2:1 to receive Rocapuldencel-T plus sunitinib versus standard of care. Even if immune responses were recorded, the trial failed its primary endpoint of improving OS compared to the control group (mOS 27.7 vs. 32.4 mos; HR = 1.10, 95% CI, 0.83–1.40). Still, a trend toward better OS was evidenced in the case of more robust immune responses [94]. In the adjuvant setting, autologous-antigens loaded DCs plus cytokine-induced killer cells (CIK) were compared to α-interferon (IFN) in 410 patients, improving PFS and OS (3-year OS rate 96% vs. 83%; 5-year OS rate 96% vs. 74%; *p* < 0.01) [95].

Among the peptide vaccines, EC90, a folate-targeted vaccine, plus α-IFN and IL-2, induced seven SDs and one PR in 24 patients in a phase I/II study [96]. Nine patients with progressive mRCC, treated with hypoxia-inducible protein-2 (HIG-2) peptide vaccine obtained a DCR of 77.8% and an mPFS of 10.3 mos [97]. GX301 vaccine is composed by four telomerase peptides plus Imiquimod and Montanide ISA-51 as adjuvant [98]. Telomerase contributes to tumor immortalization, but it is not expressed by somatic cells [99]. GX301 induced specific immunological responses in over 2/3 of vaccinated mRCC or mCRPC patients, with a trend for better OS (around 11 mos) [98].

Among the different subtypes of renal cancer, clear-cell renal cell carcinoma (ccRCC) is exceptionally responsive to immunotherapy and has been historically considered the ideal subtype to treat with vaccines [100]. However, in the IMPRINT phase III trial, IMA901 (composed of 10 tumors-associated peptides) plus GM-CSF, cyclophosphamide and sunitinib did not improve OS for the 339 randomized patients in the first-line setting (mOS 33.2 vs. 33.7 mos; HR = 1.34, 95% CI, 0.96–1.86; *p* = 0.087), even if immune activation had previously been evidenced in phase II [101,102]. PPV with vascular endothelial growth factor receptor (VEGFR)-1 was administered in 18 ccRCC patients. Among them, two PRs and five SDs with a median duration of response of 16.5 mos were observed [103]. TG-4010 is an MVA vector-based vaccine of IL-2 and MUC-1 that induced an mOS of 19.3 mos among the 27 ccRCC patients in a phase II trial [104].

Aiming to identify predictive biomarkers for therapeutic vaccines, blood parameters at baseline (platelets, neutrophils, monocytes, hemoglobin and LDH), the presence of bone metastases, the MSKCC score, the Fuhrman grade and the ECOG-performance status have been investigated in RCC [66,68,95].

Novel vaccine targets have been proposed for future clinical studies: hypoxia-inducible factor (HIF)-1α, being the RCC often associated with the mutation of Von Hippel-Lindau (VHL) gene and dependent on the upregulation of HIF; PD-L1 derived peptides, as RCC is sensitive to immunotherapy control [105,106]. Among the ongoing trials, NCT02950766 evaluates the neo-antigen NeoVax plus Ipilimumab (phase I); the NCT03289962 phase I study is testing the vaccine RO7198457 plus atezolizumab; the NCT03294083 phase Ib trial is assessing the Pexa-Vec vaccine (Thymidine Kinase-Deactivated Vaccinia Virus) plus the anti-PD1 Cemiplimab. Finally, the NCT02643303 phase I/II study evaluates the combination of tremelimumab as in situ vaccination, durvalumab and the TME modulator polyICLC, in subjects with advanced solid tumors, including RCC, UC, PCa and testicular cancer (Table 2).

## 4. Discussion

In the last years, immunotherapy has widened the therapeutic scenario of many cancer subtypes. The top results have been obtained in the urological field among RCC and UC. In the mRCC, ICIs prolonged survival as single agents in pretreated patients and combinations with other ICIs or TKIs in the first line [1,2,3,4,5,6,7]. In the mUC, ICIs were superior to chemotherapy in the second-line setting; in the first line, pembrolizumab and atezolizumab showed superiority to chemotherapy for PD-L1 positive cisplatin-unfit patients, whereas the combination of ICIs and chemotherapy did not confer a significant survival advantage [8,9,10,11,12]. The anti-PD-L1 avelumab prolonged OS as maintenance therapy in mUC [13]. Regarding PCa, only minimal efficacy has been detected, mainly due to its immunosuppressive TME; therefore, to potentiate the immune system stimulation, novel combinations with chemotherapy, TKIs and Poly ADP-ribose polymerase (PARP)-inhibitors are currently under evaluation [14,15,16,17,18,19].

However, besides blocking the immune-suppressive signals that decrease the antitumor response, the other way to enhance the immune system’s antitumor activity is the active stimulation against specific TAAs. In fact, TAAs on tumor cells, after binding with the MHC molecules on the APCs, induce T-cell activation and specific immune response. However, this theoretically effective mechanism has led to a small number of responders in most studies, with no significant survival improvements, even among those developing specific immune responses (Table 1). Different points of view could explain this failure. For example, various TAAs are expressed in different tumor areas because of tumor heterogeneity, many of which could not be targeted by the administered vaccine. Furthermore, not all the peptides can induce specific immune responses, and the selection of immunodominant peptides represents another limitation of vaccines. In fact, the frequency of a TAA does not always relate to its immunogenicity; therefore, in some cases, a weak immune response is activated. An example of this concept emerged from a retrospective study of mCRPC and mUC patients treated with PPV, in which different antigens selection determined different survival values [107]. Finally, immune escape mechanisms can develop after vaccination, leading to its failure [108]. In vaccine therapy, immune escape relies on the exhaustion of effector T-cells due to the upregulation of inhibitory molecules, such as PD-1, CTLA-4 and Lymphocyte-activation gene (LAG)-3, on T-cells surfaces. The immune escape is favored by the continuous exposure to antigens and potentiated by different elements within the TME, such as cells (Tregs, myeloid-derived suppressor cells (MDSCs), tumor-associated macrophages (TAMs) and natural killer (NK) cells), cytokines (γ-IFN, IL6, IL10 and TGF-β) and other soluble factors (Indoleamine-pyrrole 2,3-dioxygenase (IDO) and VEGF), and determines T-cells loss of function [109,110] (Figure 2).

This weakness could rationally be overcome by combining vaccines and ICIs, resulting in a more robust T-cells activation [111]. Effectively, vaccines induce specific T-cells infiltration that makes the TME more susceptible to ICIs, and conversely, ICIs interrupt the inhibitory effect of the PD-1/PD-L1 pathway [112]. Indeed, the concomitant or sequential administration of vaccines and ICIs remains an open question, together with the time and administering schedule [110]. Of note, the immunosuppressive elements within the TME, such as Tregs, MDSCs, TAMs, NK cells, TGF-β, IL-10, IDO and VEGF, are also a future possibility for combining vaccine therapy with other agents directed specifically against these pathways [30].

Besides the ICIs combinations, we should investigate other possible options for potentiating the vaccines’ efficacy. For example, it is well-known that RT has immunomodulant effects, i.e., increased MHC expression, APCs activity and inflammatory cytokine production; nevertheless, the combination of vaccines plus RT regimens has been poorly investigated after the first experiences reporting no significant results [113,114]. Moreover, the combination of vaccines and chemotherapy has been inadequately tested, as for many years the incompatibility of these two pharmacological classes has been postulated, and some trials were prematurely stopped without reaching the accrual or even due to an increased death rate [69,71,115,116]. Regarding TKIs, it is known that they exhibit immunomodulatory properties, such as increased tumor infiltration of T-cells and reduction of the production of anti-inflammatory mediators, explaining the success of combinations with ICIs also among GU tumors [1,2,3,4,5,117]. However, TKIs also exert immunosuppressive properties, such as reducing the production and function of T/NK cells and inducing the production of IL-10 [117]. Immunosuppressive effects have also been reported when TKIs were concomitantly administered with vaccines, whereas a potentiation of immune responses was evidenced after sequential therapy [118]. Therefore, future new combinations with vaccines, dosage and, above all, timing, should be carefully investigated.

Despite many years of experience with vaccines, most studies did not get over the I/II phases and limited numbers of treated patients. Therefore, randomized trials enrolling larger populations could clarify the possible advantages and future applications in daily practice. In addition to them, preclinical studies, including cancer cell cultures and animal models, could further elucidate the vaccines’ role in the GU field, allowing a more profound knowledge of tumor immunologic features and their interactions for dosage, timing and combinations to be optimized. Furthermore, it is of the utmost importance to search for tissue and blood biomarkers with a predictive role to better perform a more accurate patient selection [119,120,121].

## 5. Conclusions

Immunotherapy is quickly changing the treatment landscape of many solid tumors, including those of the GU tract. However, low response rates and resistance mechanisms force new alternative pathways to be explored. Vaccines might represent effective means to stimulate the immune system. Their concomitant or sequential combination with other regimens must be more deeply evaluated in prospective trials to maximize the benefits for the highest number of patients. More extensive knowledge of the biological and immunological features of the different malignancies and the interactions with the host immune system is needed. Biomarkers with predictive roles will improve patient selection and enhance survival outcomes in the GU field.

## Figures and Tables

**Figure 1 vaccines-09-00623-f001:**
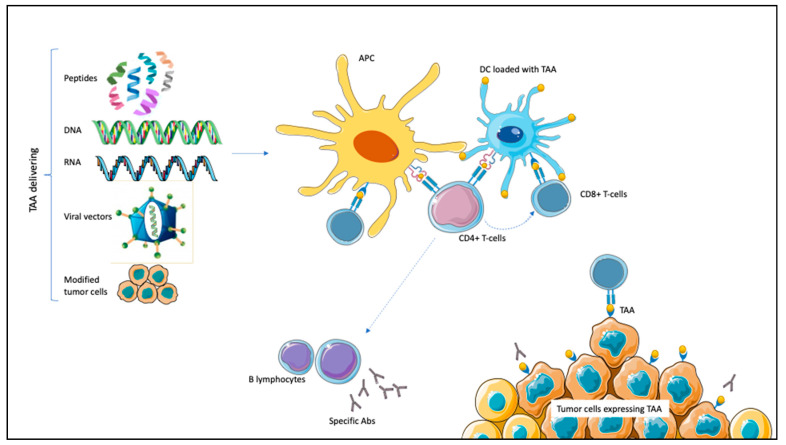
Mechanism of action of therapeutic vaccines in genitourinary malignancies. Tumor-associated antigens (TAAs) are expressed on tumor cells. TAAs can be delivered by different mechanisms (peptides, DNA or RNA encoding TAAs, and viral vectors carrying TAAs or modified tumor cells), leading to antigen-presenting cells (APCs) activation. Dendritic cells (DCs) are themselves APCs and can be loaded with TAAs. After antigen processing, APCs interact with CD8^+^ T-cells through class I major histocompatibility complex (MHC), inducing specific cytotoxic responses against TAA-expressing tumor cells. Through class II MHC, APCs activate CD4^+^ T-cells. CD4^+^ potentiate CD8^+^ activation; moreover, they induce B-lymphocytes activation for specific antibodies (Abs) production against TAA-expressing tumor cells.

**Figure 2 vaccines-09-00623-f002:**
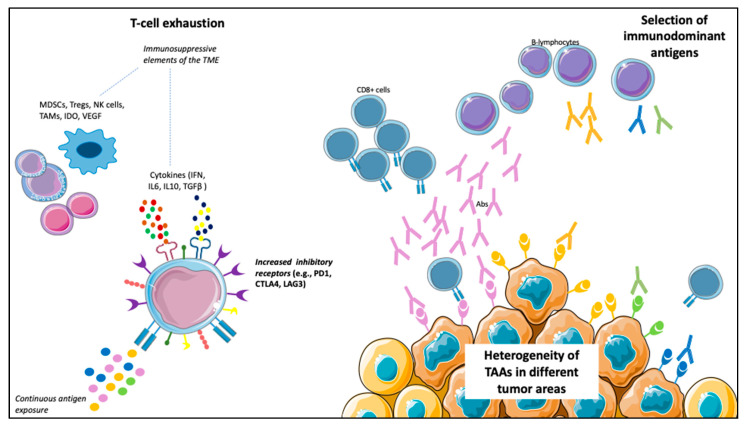
Possible mechanisms limiting vaccines’ efficacy. (1) T-cells exhaustion: the continuous antigen exposure, together with cells (e.g., myeloid-derived suppressor cells (MDSCs), tumor-associated macrophages (TAMs), natural killer (NK) cells and Treg cells), cytokines (e.g., interferon (IFN), interleukin (IL)-6, IL10, transforming growth factor (TGF)-β) and other soluble factors (e.g., vascular endothelial growth factor (VEGF) and indoleamine-pyrrole 2,3-dioxygenase (IDO)), results in loss of function of T-cells, with an increased expression of inhibitory receptors (e.g., programmed death (PD)-1, cytotoxic T-lymphocyte antigen (CTLA)-4, lymphocyte-activation gene (LAG)-3) on T-cells surface. (2) Heterogeneity of tumor-associated antigens (TAAs) in different tumor areas: a particular vaccine could not target all the tumor areas. (3) Selection of immunodominant antigens capable of triggering a robust immune response (of note, the frequency of an antigen does not correlate with its immunogenicity).

**Table 1 vaccines-09-00623-t001:** Vaccine therapies in genitourinary malignancies. Principal TAAs and key findings of the studies with therapeutic cancer vaccines are reported.

TAA	Vaccine Name	Type of Vaccine	Combination	Population	Phase	Key Findings	Reference
PAP	Sipuleucel-T (Provenge^®^)	DC	/	mCRPC	III	mOS: 25.8 vs. 21.7 mos (HR = 0.78; 95% CI, 0.61–0.98; *p* = 0.03); no PFS improvement; lower baseline PSA levels predictive of OS	[26,31]
			ADT	nmCRPC	II	Humoral response with Sipuleucel-T→ADT than vice versa, related to longer TTP for PSA (*p* = 0.007)	[32]
			Abiraterone	mCRPC	II	Immune responses, not reduced by prednisone	[33]
			Ipilimumab	mCRPC	I	>4 years OS in 6/9 pts	[34,35]
			/	Neoadjuvant PCa	II	T-cells activation in tumor biopsies	[36]
	pTVG-HP	DNA	/	mCRPC		PSA decline in ~60% patients	[37]
			Pembrolizumab	Recurrent PCa	II	No MFS improvement	[38]
PSA	PROSTVAC (PSA-TRICOM)	Viral vector	/	mCRPC	III	No survival improvement; early terminated	[40]
			/(intraprostatic)	Recurrent PCa	I	Increased CD4^+^/CD8^+^ in tumor biopsies, PSA SD in 10/19 pts	[41,42]
			Ipilimumab	mCRPC	I	PSA decline in ~50% pts, low PD1^+^ /high CTLA4^−^ Tregs associated with longer OS	[43,44]
PSMA		DNA	/	nmCRPC	I/II	PSA-DT 16.8 vs. 12.0 mos (*p* = 0.0417)	[45]
		VRP	/	mCRPC	I	Antibodies production; no clinical benefit	[48]
PSA + PSMA	INP-5150	DNA	/	nmCRPC	I/II	18 mos PFS rate: 85%	[46]
PSMA + Survivin		DC	(vs. Docetaxel + prednisone)	mCRPC	I	ORR: 72.7% vs. 45.4%	[47]
PSMA + PS + PSCA + STEAP1	CV-9103	RNA	/	mCRPC	I/II	Immune responses	[49]
AR	pTGV-AR	DNA	/	mHSPC	I	Longer PSA-PFS in case of T-cells activation (*p* = 0.003)	[51]
MUC1		DC	/	nmCRPC	I/II	Improved PSA-DT (*p* = 0.037)	[52]
MUC1 + PSA + Brachyury		Viral	/	mCRPC	I	PSA decline in 2/12 pts	[53]
MUC1 + IL2	TG-4010	Viral vector	/	ccRCC	II	mOS: 19.3 mos	[104]
NY-ESO-1		Peptide	/	Stage IV PCa	I	T-cell responses in 9/12 pts, no survival data	[55]
NY-ESO-1 + MAGE-C2 + MUC1		DC	/	mCRPC	IIa	T-cell responses in ~30% pts, related to radiological responses	[56]
HER-2	AE37	Peptide	/	HER-2^+^ PCa	I	Long memory (4 years) with multiple boosters; pre-existing immunity related to PFS, TGF-β inversely related to OS, HLA-A*24/DRB1*11 related to OS	[57,58,59,60]
CDCA1		Peptide	/	mCRPC	I	mOS: 11 mos	[61]
UV1		Peptide	/	mHSPC	I	Immune responses in 85.7%, PSA declining in 64% pts	[62]
TARP		Peptide + DC	/	D0 PCa	I	Specific immune responses, reduced PSA velocity	[63]
RhoC		Peptide	/	PCa after RP	I/II	CD4+ responses in 18/21 pts	[64]
5T4		Double viral vector	/	Neoadjuvant, active surveillance—PCa	I	T-cell responses before RP and during active surveillance	[65]
	TroVax	Viral	Docetaxel	mCRPC	II	mPFS: 9.67 mos (vs. 5.1 docetaxel alone; *p* = 0.097), related to baseline PSA	[68,69]
Modified PCa cells	GVAX	Cell line	Docetaxel	Neoadjuvant PCa	II	Gleason score downstaging in 4/6 pts	[70,71]
			Degarelix + cyclo-phosphamide	Neoadjuvant PCa	I/II	Immune responses	[72]
PPV		Peptide	/	mCRPC	III	No survival advantage (HR = 1.04; *p* = 0.77); OS benefit with very low/high baseline lymphocytes	[74,75,76,77]
	DCvac	DC	Docetaxel	mCRPC	II	Immune responses, no survival advantage	[80]
		Peptide	/	BCa	I	mOS: 7.9 mos (vs. 4.1 BSC; *p* = 0.049), no PFS advantage	[91]
			Alone or plus chemotherapy	mUTUC	II	Longer OS in case of immune response (*p* = 0.019); mOS: 7.7 mos (13.0 mos plus CT);	[92]
		Peptide	/	mUC	I	1/12 CR, 1/12 PR, 2/12 SD, mPFS 3 mos, mOS 8.9 mos	[93]
20-peptides	KRM-20	Peptide	/	mCRPC	I	2/17 PR, 1/17 PSA stability	[81]
			Docetaxel + dexamethasone	mCRPC	II	Increased specific antibodies and T-cells, no PSA/OS differences vs. PBO	[82]
MAGE-A3		Peptide	Before BCG	NMIBC	I	Specific T-cells in ~50% pts, no survival data	[85]
Survivin		Peptide	/	mUC	I, II	Improved OS (*p* = 0.0009)	[86]
Mannose receptor	CDX-1307	Peptide	/	mUC	I	Immune responses, early stopping of phase II due to slow enrollment	[87]
WT1		DC	/	mUC, mRCC	I/II	Specific immune responses, decreased Tregs	[88]
DEPDC1 + MPHOSPH1	S-288310	Double peptide	/	mUC	I/II	mOS: 14.4 mos, better results with immune response against two peptides	[89]
NEO-PV-01		Peptide	/	BCa	Ib	PR/SD in 10/14 pts	[90]
CD40L + RCC RNA	Rocapuldencel-T	DC + RNA	Sunitinib	mRCC	III	No OS improvement over Sunitinib (mOS 27.7 vs. 32.4 mos; HR = 1.1, 95% ci, 0.83–1.40); trend for better OS in case of robust immune response	[94]
Autologous antigens		DC	CIK	Resected RCC	III	Compared to α-IFN, PFS improvement; 3-year OS rate 96% vs. 83%; 5-year OS rate 96% vs. 74%; *p* < 0.01	[95]
Folate	EC-90	Peptide	α-IFN, IL-2	mRCC	I/II	7/24 SD, 1/24 PR	[96]
HIG-2		Peptide	/	mRCC	I	DCR 77.8%, mPFS 10.3 mos	[97]
Telomerase	GX301	Peptide	/	mRCC, mCRPC	I/II	Immune responses with trend for better OS (~11 mos)	[98]
10-peptides	IMA901	Peptide	Sunitinib	ccRCC	III	No OS advantage (mOS 33.2 vs. 33.7 mos; HR = 1.34, 95% CI, 0.96–1.86; *p* = 0.087)	[101,102]
VEGFR1		Peptide	/	ccRCC	I	2/18 PR, 5/18 SD, mDOR 16.5 mos	[103]

AR, androgen receptor; BCa, bladder cancer; ccRCC, clear-cell renal cell carcinoma; CDCA1, cell division associated 1; CI, confidence interval; CIK, cytokine-induced killer cells; CR, complete response; CT, chemotherapy; DC, dendritic cells; DCR, disease-control rate; DEPDC1, DEP domain-containing 1; GM-CSF, granulocyte–macrophage colony-stimulating factor; HER-2, human epidermal growth factor receptor 2; HIG-2, hypoxia-inducible protein 2; HR, hazard ratio; IFN, interferon; IL, interleukin; MAGE, melanoma-associated antigen; mCRPC, metastatic castration resistant prostate cancer; mDOR, median duration of response; MFS, metastasis-free survival; mHSPC, metastatic hormone sensitive prostate cancer; mOS, median overall survival; m PFS, median progression-free survival; MPHOSPH1, M-phase phosphoprotein 1; mRCC, metastatic renal cell cancer; mUC, metastatic urothelial cancer; MUC1, mucin-1; mUTUC, metastatic upper tract urothelial cancer; nmCRPC, non-metastatic castration resistant prostate cancer; NMIBC, non-muscle invasive bladder cancer; ORR, overall response rate; PAP, prostatic acid phosphatase; PBO, placebo; PCa, prostate cancer; PPV, personalized peptide vaccination; PR, partial response; PSA, prostate specific antigen; PSA-DT, PSA doubling time; PSCA, prostate stem cell antigen; RhoC, Ras homolog gene family member C; RP, radical prostatectomy; SD, stable disease; STEAP1, six-transmembrane epithelial antigen of the prostate-1;TAA, tumor-associated antigens; TARP, T-cell receptor gamma chain alternate reading frame protein; TGF, transforming growth factor; TTP, time to progression; VEGFR, vascular endothelial growth factor receptor; VRP, viral replicon vector; WT, Wilms tumor.

**Table 2 vaccines-09-00623-t002:** Ongoing trials with therapeutic vaccines and their combinations in genitourinary malignancies.

**Clinicaltrials.gov Registration Number**	**Phase**	**Setting**	**Vaccine**	**Antigen**	**Combination**
NCT01804465	II	mCRPC	Sipuleucel-T	PAP	Ipilimumab (immediate vs. delayed)
NCT02463799	II	mCRPC	Sipuleucel-T	PAP	Radium-223
NCT01881867	II	mCRPC	Sipuleucel-T	PAP	Glycosylated recombinant human IL-7
NCT03600350	II	nmCRPC	pTGV-HP	PAP	Nivolumab
NCT04090528	II	mCRPC	pTGV-HP + pTGV-AR	PAP, AR	Pembrolizumab
NCT02933255	I/II	NAD PCa	PROSTVAC	PSA	Nivolumab
NCT03532217	I	mHSPC	PROSTVAC	PSA	Neoantigen DNA vaccine, Nivolumab, Ipilimumab
NCT02649855	II	mHSPC	PROSTVAC	PSA	Docetaxel
NCT03315871	II	nmCPRC	PROSTVAC	PSA	M7824 (anti-PD-L1/TGF-βR2), CV301
NCT02325557 (KEYNOTE-146)	I/II	mCRPC	ADX31-142	PSA	Pembrolizumab
NCT02362451	II	nmCRPC	TARP DC	TARP	/
NCT02111577	III	mCRPC	DCvac	PPV	Docetaxel vs. PBO
NCT03412786	I	mHSPC	Bcl-xl_42-CAF09b peptide vaccine	BCl-xl	/
NCT04701021	I	Relapsing PCa after RP	TENDU peptide conjugate	TET	/
NCT04114825	II	Biochemical recurrent PCa after RT/RP	RV001V peptide vaccine	RhoC	/
NCT03493945	I/II	mCRPC	BN-Brachyury	Brachyury	M7824, ALT-803, Epacadostat
NCT03689192	I	mUC	ARG1	Arginase-1	/
NCT03715985	I/II	mUC	NeoPepVac	Personalized neoantigen	Anti-PD1/PD-L1
NCT02950766	I	mRCC	NeoVax	Personalized neoantigen	Ipilimumab
NCT03289962	I	mRCC	RO7198457	20 TAAs	Atezolizumab
NCT03294083	Ib	mRCC	Pexa-Vec	Thymidine-kinase	Cemiplimab
NCT02643303	I/II	Advanced RCC, UC, PCa, testicular cancer	In situ vaccination with tremelimumab		Durvalumab, polyICLC

AR, androgen receptor; BCl-xl, B-cell lymphoma extra-large protein; IL, interleukin; mCRPC, metastatic castration resistant prostate cancer; mHSPC, metastatic hormone sensitive prostate cancer; mRCC, metastatic renal cell carcinoma; mUC, metastatic urothelial carcinoma; NAD, neo-adjuvant; nmCRPC, non-metastatic castration resistant prostate cancer; PAP, prostatic acid phosphatase; PCa, prostate cancer; PD1, programmed death 1; PD-L1, programmed death-ligand 1; PPV, personalized peptide vaccination; PSA, prostate specific antigen; RhoC, Ras homolog gene family member C; RP, radical prostatectomy; RT, radiotherapy; TAA, tumor-associated antigen; TARP, T-cell receptor gamma chain alternate reading frame protein; TET, tetanus-epitope targeting; TGF, transforming growth factor.

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
