# Peer review of "Cancer Vaccines for Genitourinary Tumors: Recent Progresses and Future Possibilities"

_vaccines, 2021, doi:10.3390/vaccines9060623_

Round 1
Reviewer 1 Report
The article by BA Maiorano et al is a review of the literature on antitumor vaccines for urogenital tumors. The review presents in the first chapter a comprehensive list of completed or ongoing clinical trials of vaccines. The second chapter is a discussion that puts into perspective the questions that will need to be answered to improve vaccine efficacy.
This review is very didactic and I enjoyed reading it.
I have no major comments.
Minor comments.
In table 2, maybe add the name of the vaccine antigen when available.
As a review is often an opportunity to schematize mechanisms/processes, the authors should add a figure presenting schematically the immunosuppression processes that limit the efficacy of antitumor vaccines (mirroring figure 1 on the vaccination process).
The authors should mention in the introduction the difficulties of generating cytotoxic responses with extracellular antigens and must add a short paragraph on the process of antigen cross-presentation. Cite 1 or 2 reviews that detail this essential process that has revolutionized antitumor vaccination strategies.
Author Response
We thank the reviewer for the constructive comments.
We added the available antigen names of the vaccines under study in Table 2.
We built a figure (Fig.2) schematizing the possible mechanisms limiting the efficacy of anti-tumor vaccines.
We summarized the activity of the antigen-presenting cells and mentioned the antigen cross-presentation adding a paragraph in the Introduction section.
Reviewer 2 Report
General comment: The authors presented an interesting review work concerning to the cancer vaccines for genitourinary tumors.
The manuscript is already published in the following website: https://www.preprints.org/manuscript/202105.0205/v1
In a general way, the manuscript is well written.
Title: It is clear, concise, and adequate.
Introduction: It is adequate.
Materials and Methods: They are adequate.
Results: They are clearly described and supported by the Figures and the Tables.
Discussion and conclusions: It is adequate.
Recommendation: If clarified the publication of the study in the website: https://www.preprints.org/manuscript/202105.0205/v1, it should be accepted for publication.
Author Response
We thank the reviewer for the positive comments.
Regarding the preprints.org website, the MDPI system provided the Preprints option to make a not peer-reviewed version of the manuscript available online immediately.
Reviewer 3 Report
This is a review about use of anti-cancer vaccines in urogenital cancer. The review appears to be as comprehensive as possible, perhaps relying a little too much on phase 1 trials. Although inclusion of phase 1 work does preview some upcoming phase 2 and 3 trials, may prevent duplication of effort or encourage new combinations to be considered.
In section 2, last sentence, there is mention of inclusion criteria. No such criteria are otherwise mentioned. Please, list the criteria.
In Table 1, statistics are not provided in most cases for parameters beyond OS or PFS. This appears to be sometimes mentioned in the text . Can some way be found to enter this into the Table or indicate that the information is in the text (* as a footnote). This also applies to the first listed Phase3 trial-mOS; was this significant? Given the few number of phase 3 trials and the apparent lack of success at this stage, clarification is important. Midway through Table 1 (key was reference 70), the study is missing phase classification.
Minor-the phrase-"data are [not] disposable" does not make sense, although I can guess what it means.
Author Response
We thank the reviewer for the comments and the suggestions for improving the manuscript.
We listed inclusion/exclusion criteria in the Materials and Methods section.
In Table 1, we added more study results in the section ‘Key findings’, the missing study phases, and the requested information about the significance of phase 3 studies results.
We modified the incorrect sentences as indicated.